# DEPrune: Depth-wise Separable Convolution Pruning for Maximizing GPU Parallelism

**Cheonjun Park**[1], **Mincheol Park**[23], **Hyunchan Moon**[4], **Myung Kuk Yoon**[5],
**Seokjin Go**[6], **Suhyun Kim**[3*], **Won Woo Ro**[2*]

[1] Samsung Electronics    [2] Yonsei University    [3] Korea Institute of Science and Technology
[4] LG Electronics    [5] Ewha Womans University    [6] Georgia Institute of Technology
{cheonjun.park, mincheol.park, wro}@yonsei.ac.kr,
{mhcqwe92, dr.suhyun.kim}@gmail.com,
myungkuk.yoon@ewha.ac.kr, seokjin.go@gatech.edu

## Abstract

Depth-wise Separable Convolution (DSConv) has a powerful representation even with fewer parameters and computation, leading to its adoption by almost all of the state-of-the-art CNN models. DSConv models are already compact making it hard to apply pruning, and there are few previous pruning techniques that target depth-wise convolution (DW-conv). In this paper, we present Depth-wise Separable Convolution Pruning (DEPrune), a novel pruning method applied to both point-wise and depth-wise convolutions. DEPrune is optimized by analyzing the computation of DSConv on GPUs. DEPrune employs a fine-grained pruning approach, yet it achieves the structured sparsity typically absent in fine-grained pruning, enabling practical hardware acceleration. Moreover, this method maintains a high pruning ratio without causing any accuracy drop. We additionally represent techniques that further enhance DEPrune performance: 1) balanced workload tuning (BWT), and 2) hardware-aware sparsity recalibration (HSR). Experiment results show that DEPrune achieves up to $3.74\times$ practical speedup in DSConv inference on GPUs while maintaining the accuracy of EfficientNet-B0 on ImageNet.

## 1 Introduction

In computer vision tasks, Convolutional Neural Networks (CNNs) have dramatically gained parameters and computation [54] to solve complex and varied tasks [8]. Such massive computation and memory footprint poses challenges in environments with limited hardware resources, such as mobile devices. Many research efforts have been made to address such problems, and two techniques have been most effective: Depth-wise Separable Convolution (DSConv) [6] and DNN pruning [17].

DSConv [6, 43, 45] is composed of Depth-wise Convolution (DW-conv) and Point-wise Convolution (PW-conv), allowing it to have a similar representation power to traditional CNNs that use standard convolution, even with fewer parameters and computation [3]. Therefore, modern CNNs primarily adopt DSConv when designing models [9, 40, 46, 48].

DNN pruning eliminates redundant weight parameters without compromising representation power. Weight pruning [18] brings a significant pruning ratio ($PR$) due to the fine-grained approach but rarely reduces inference time compared to the unpruned model because of index computation overhead [54]. In contrast, structured pruning [16, 30, 32, 21, 38] is a coarse-grained approach that is GPU-friendly and leads to a practical reduction in inference time.

---

*co-corresponding authors.

38th Conference on Neural Information Processing Systems (NeurIPS 2024).

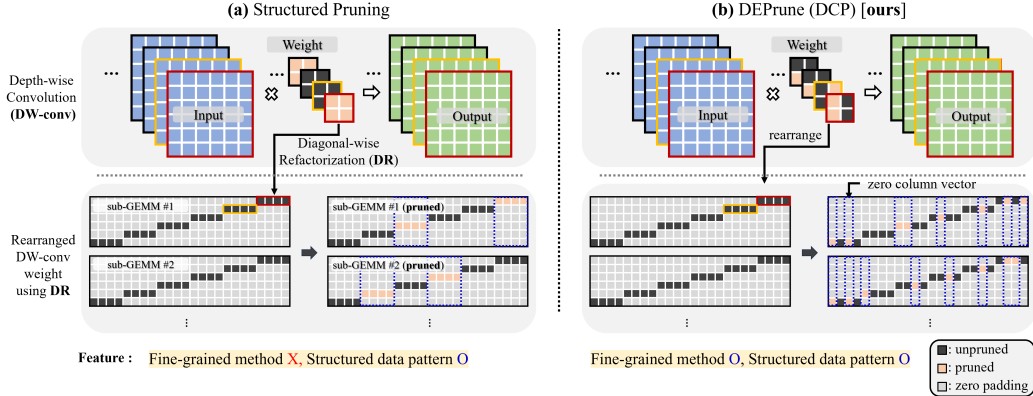

Figure 1: Depth-wise convolution is rearranged to multi sub-GEMM on GPU by applying Diagonal-wise Refactorization (DR). The 'X' and 'O' symbols indicate the absence and presence of corresponding characteristics for each method. Applying (a) Structured Pruning and (b) DEPrune (DCP) to multi sub-GEMM results in a structured data pattern. But (b) DEPrune (DCP) is more fine-grained method than (a) Structured Pruning.

In DSConv, despite the DW-conv has only about $1\%$ of the parameters, it spends over $82\%$ of the overall inference time [41]. As a result, reducing the computation time of DW-conv can inherently impact the overall network execution time, and employing pruning on DW-conv can provide an effective solution to accelerate DSConv. Applying existing structured pruning [27, 20, 37] to PW-conv is not difficult, because PW-conv has the same GPU operation characteristic as standard convolution. Conversely, DW-conv has two challenges in applying previous pruning methods. First, DW-conv has particularly fewer parameters, so applying coarse-grained pruning that creates a hardware-friendly structured data pattern causes significant accuracy loss. Second, the operation of DW-conv on GPU underutilizes the parallelism since the input unit is smaller than the operation unit, thus applying previous pruning methods is useless in this condition. Therefore, we propose **Depth-wise Separable Convolution Pruning (DEPrune)**, a hardware-aware pruning approach specialized for DW-conv for fast and memory-efficient DSConv.

First, to address the aforementioned challenges: accuracy loss and underutilization, we analyze the operation of DW-conv on a widely used GPU. On GPUs, DW-conv computes by being transformed into multiple-GEMV (GEneral Matrix-Vector Multiplication) [41], and this structure does not fully utilize the GPU parallelism. Thus, for efficient operation on GPU, previous work applies Diagonal-wise Refactorization (DR) [41]. DR is a method of rearranging DW-conv to multiple sub-GEMM (GEneral Matrix-Matrix Multiplication) operations to maximize GPU parallelism (Fig. 1). DR places the weights diagonally and zero padding the rest. Therefore, if one non-zero weight is removed, all elements on the same column line have zero value, so the corresponding line is all zero vector (Fig. 1). At this point, we apply fine-grained pruning on DW-conv rearranged by DR, as we call the **Depth-wise Convolution Pruning (DCP)** (Sec. 4.1). DCP's fine-grained approach provides novel $PR$ and since most of the sub-GEMM is zero-padded, it brings regular sparsity even with fine-grained pruning. This hardware-friendly format results in inference speedup without representation power loss. DEPrune also applies conventional structured pruning to PW-conv; however, taking into account the computational significance of the DSConv model, we selectively prune PW-conv layers to maximize the pruning ratio without sacrificing accuracy.

Second, for DEPrune enhancement we consider the overall operation flow of DSConv to optimize GPU utilization. When DCP is applied, the $PR$ is different for each sub-GEMM, which executes in different processing units. This results in a workload imbalance problem between processing units which directs to GPU under-utilization. The total execution time is set to the longest GEMM, thereby other idle processing units are forced to wait until the longest GEMM finishes. To solve this problem, we propose a **Balanced Workload Tuning (BWT)** that sets the same target $PR$ for each sub-GEMM when applying DCP (Sec. 5.1). Our DEPrune applies existing structured pruning [37, 27, 20] on PW-conv, and this approach avoids workload imbalance problem.

|  | Pruning for **DSConv** | | | | |
|  | Pruning for **DW-conv** | | | Pruning for **PW-conv** | |
|  | DCP | Enhance Method | | Structured† | Enhance Method |
| Status | [**ours**] | BWT [**ours**] | HSR [**ours**] | Pruning | HSR [**ours**] |
| DEPrune | ✓ | - | - | ✓ | - |
| DEPrune-B | ✓ | ✓ | - | ✓ | - |
| DEPrune-BH | ✓ | ✓ | ✓ | ✓ | ✓ |

Table 1: Terminology of DEPrune method. This symbol (†) means 'we apply our methodology to determine which PW-conv to prune for better performance (Sec. 4.2)'. BWT and HSR are our proposed method to enhance DEPrune. BWT and HSR are described in Sec. 5.1 and Sec. 5.2, respectively. This symbol (✓) means 'Applied'.

Lastly, for DEPrune enhancement we found that in addition to balancing $PR$ between sub-GEMMs, adjusting the pruning ratio proportional to the GPU execution unit could further lead to significant speed improvements. Due to our technique's structured data format, theoretically, the acceleration should increase proportionally as the pruning ratio increases. However, because GPU operates in a specific execution unit, the computational workload should scale with the unit of execution to maximize the acceleration effect. Considering this unit operation, we additionally determine negligible weight parameters for more pruning, and this results in an additional significant speed improvement of 3-16% for DW-conv and also PW-conv without any accuracy loss. We call this **Hardware-aware Sparsity Recalibration (HSR)**, an algorithm that recalibrates the appropriate target $PR$ for DW-conv and PW-conv considering the execution unit of the GPU (Sec. 5.2).

To further optimize DSConv, various techniques are considered, which led to a diverse and somewhat complex set of terms being used throughout the paper. Therefore we provide Table 1 which summarizes the structure and terminology of our DEPrune.

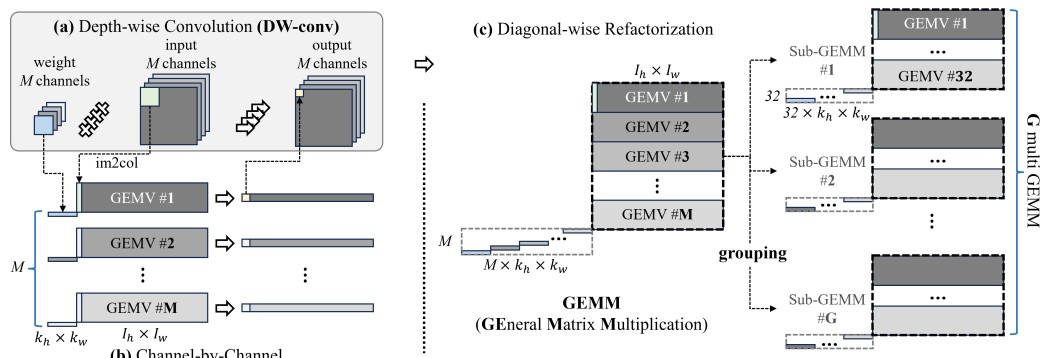

Figure 2: (a) DW-conv is rearranged to multi GEMV through (b) Channel-by-Channel on GPU execution. (c) Diagonal-wise Refactorization (DR) rearranges DW-conv into multiple sub-GEMMs. After DR, due to GPU tile size [14], we group **M** GEMVs into units of **32**, resulting in a total of **G** sub-GEMMs.

## 2 Preliminary

**Prerequisites**  DW-conv's weight filter is 3D tensor. Given $l^{th}$ DW-conv layer as $\mathcal{D}^{(l)} \in \mathbb{R}^{M \times k_h \times k_w}$, where $M$, $k_h$, and $k_w$ are the number of channels, height, and width of the filters, respectively. $I_h$, and $I_w$ are the height and width of the input, respectively.

**Channel-by-Channel**  As shown in Fig. 2-(a), DW-conv is composed of 3D input ($\mathbb{R}^{M \times I_h \times I_w}$), 3D Weight ($\mathbb{R}^{M \times k_h \times k_w}$), each with M channels performing independent 2D convolution operations. GPU rearranges standard convolution to GEMM, using im2col [2, 5] to enable data reuse. Similarly, major deep learning frameworks (e.g., Caffe, PyTorch, MXNet, and TensorFlow) rearrange DW-conv to $M$ multiple GEMV operations, using Channel-by-Channel (Fig. 2-(b)). GEMV consists of a

weight vector of size $k_h \times k_w$ and an input matrix of size $(k_h \times k_w) \times (I_h \times I_w)$. However, this approach suffers from the limitation of weight vector size being too small (9 or 25) to fully utilize the GPU's processing units effectively.

**Diagonal-wise Refactorization (DR)**   To address under-utilization, Diagonal-wise Refactorization (DR) [41] arranges the weight vectors of the GEMVs diagonally and sequentially places the input matrix (Fig. 2-(c)). Next, zero padding is added to the empty spaces to create a complete dense GEMM, which consists of $M \times (M \times k_h \times k_w)$ weight matrix and an $(M \times k_h \times k_w) \times (I_h \times I_w)$ input matrix. However, the rearranged GEMM is excessively large $(M \times k_h \times k_w)$, so this operation requires significant additional computation on the GPU such as tiling [41]. Therefore, DR further divides this dense GEMM into smaller sub-GEMMs of a certain size. When executing matrix multiplication on GPUs, grouping with a size of 32 channels is found to be the most efficient, resulting in a total of $G$ sub-GEMM operations ($G = \frac{M}{32}$). Thus each sub-GEMM is composed of a $32 \times (32 \times k_h \times k_w)$ weight matrix and a $(32 \times k_h \times k_w) \times (I_h \times I_w)$ input matrix. This approach allows for highly optimized GPU execution using specialized cuDNN libraries [5].

# 3   Related Works

## 3.1   Hardware-aware Pruning

Among the previous DNN pruning techniques, the following three methods consider hardware characteristics to reduce inference time: structured pruning, balanced pruning, and block pruning. Structured pruning [32, 16, 53] determines redundancy at the vector level for pruning, thereby creating a regular sparsity. This structured data pattern requires almost no additional index computation on GPU, making it effective in reducing inference time [54]. Block pruning [15, 36, 49] that considers the tiling technique [5], applies structural pruning at the small matrix level and has less representation power loss at the same $PR$ compared to structured pruning. Balanced pruning [58, 51, 31, 23, 37] is a technique that divides the weight into consistent ranges and assigns an equal $PR$ to each segment, ensuring a workload balanced characteristic. Most balanced pruning achieves approximately a $2\times$ speedup when a specific GPU with a dedicated accelerator (e.g. sparse tensor core) is used at only $50\%$ $PR$ [35]. These pruning methods are based on the optimization technique of lowering for standard convolutions. Therefore, such pruning methods are difficult to apply to DW-conv. However, our DEPrune considers the hardware computation of DW-conv, enabling performance improvement.

## 3.2   Optimizations for DSConv

Since DSConv operates differently from standard convolution, continuous research is rapidly ongoing for the optimization of DSConv through dedicated software and hardware optimization.

On the pruning side, Multi-stage gradual pruning [47] prune the filters on DSConv using gradual pruning principle [57]. Probability-based channel pruning [56] considers batch normalization when pruning in DSConv and requires little fine-tuning. WP-UNet [42] utilizes fine-grained pruning in DSConv to merely reduce parameters without considering speedup. The filter pruning [34] sorts and prunes the filters according to their variance in each DW-conv. However, existing studies either only apply pruning on PW-conv or do not lead to noticeable substantial inference speed improvement even if pruning is also applied to DW-conv.

On the software side, DepthShrinker [10] removes non-linear activation functions after training and merges consecutive linear operations into a single dense operation to maximize hardware efficiency without compromising accuracy.

On the hardware side, GPU optimization [33] proposes a dynamic tile size scheme for GPUs to improve GPU utilization and hide memory access latency in DW-conv. The method [55] suggests loop rescheduling and register tiling on DW-conv, because when executing DW-conv on the parallel processor, traffic overload occurs between the cache, memory, and register. Diagonal-wise Refactorization (DR) [41] maximizes the parallelism of the GPU by proposing a rearrange method that combines all filters of DW-conv into a multi GEMM.

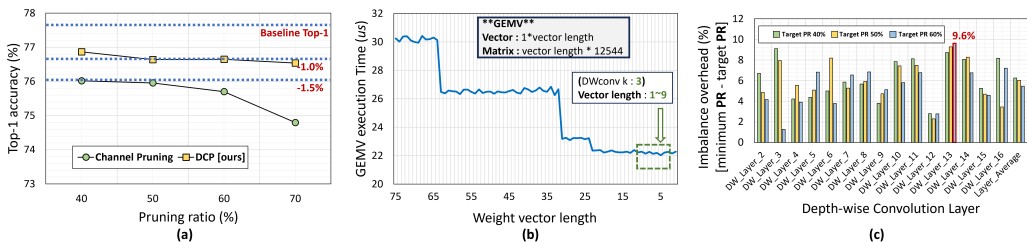

Figure 3: (a) Comparison of accuracy drop between DCP and channel pruning on EfficientNet-B0 using ImageNet. (b) Measurement of the GEMV execution time of DW-conv 6th layer of EfficientNet-B0 on GPU. (c) Measurement of imbalance overhead of Mobilenet-V2 on ImageNet. The imbalance overhead is the difference between minimum sub-GEMM pruning ratio ($PR$) and layer's target $PR$.

## 4  Proposed Method: DEPrune

### 4.1  DCP: Depth-wise Convolution Pruning

**(a) Motivation 1: Channel pruning on DW-conv has a large pruning unit size problem**    As shown in Fig. 2-(b), DW-conv generates a multi-GEMV format for each channel, on GPUs. DW-conv can also achieve structured data format, by evaluating the significance of each GEMV and eliminating an unnecessary weight vector of GEMV. Nevertheless, when compared to the 4D tensor weight of the standard convolution, the DW-conv weight is a 3D tensor ($\mathbb{R}^{M \times k_h \times k_w}$), notably fewer parameters. Consequently, eliminating a single channel ($\mathbb{R}^{1 \times k_h \times k_w}$) from DW-conv can greatly diminish its representation power. As shown in Fig. 3-(a), when pruning is done channel-wise, there's a representation power loss of $1.66\%$ compared to the unpruned model even at just a $40\%$ $PR$ (EfficientNet-B0 on ImageNet). This indicates that channel-wise pruning on DW-conv is not an appropriate choice.

**(b) Motivation 2: Hardware-unfriendly problem of weight pruning without DR**    Weight pruning experiences the least representation power loss among DNN pruning techniques with the highest pruning ratio. When applying weight pruning to the multiple GEMVs of DW-conv, the representation power loss due to increased $PR$ is much less than the previously mentioned channel pruning. Looking at Fig. 3-(a) as our DCP similar to weight pruning, there are very minor representation power losses of $0.94\%$ and $1.15\%$ at $50\%$ and $70\%$ $PR$s, respectively. However, weight pruning without considering DR does not result in practical speedup from pruning. As shown in Fig. 2-(b), the vector size of DW-conv GEMV is $k_h \times k_w$ (e.g., 9 or 25). Since this is smaller than the GPU's tile size (32), there is almost no change in inference time (Fig. 3-(b)) since GEMV underutilizes processing units of GPU.

**(c) Method: DCP**    We propose Depth-wise Convolution Pruning (DCP) to address the above two issues. We discover that weight pruning after DR can even achieve a structured sparsity in DW-conv with high $PR$, and making large matrix multiplication fully utilizes GPU parallelism. As shown in Fig. 4, first, we take the weight matrix rearranged in the form of matrix multiplication by DR. The height of the weight matrix is $M$, and the width is $M \times k_h \times k_w$. As shown in Fig. 4, the unpruned values in the weight matrix are placed diagonally, while the rest are zero-padded. Second, we sort the unpruned values in ascending order and select the threshold value that corresponds to the target pruning ratio. When calculating the target pruning ratio, zero-padded values are not considered. Last, for each unpruned value, if it is smaller than the threshold, we change it to $0$ (i.e., magnitude pruning [18]). Since the other values in the same column are already zero values, the column vector becomes a zero column vector, which is hardware-friendly.

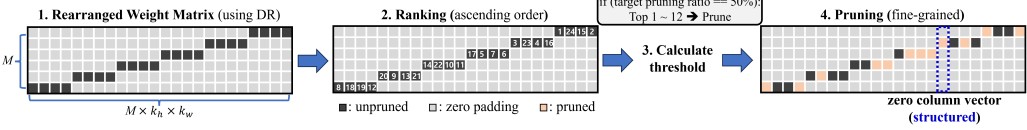

Figure 4: Process of Depth-wise Convolution Pruning (DCP).

## 4.2 Methodology for Determining which PW-conv Layer to Prune

When filter pruning is applied to PW-conv, the parameters of the subsequent layers are removed with the same sparsity. DSConv has the following structure: PW-conv1→DW-conv→PW-conv2. In DSConv, if PW-conv1 is filter pruned, the parameters of DW-conv are also removed at the channel level. The existing PW-conv pruning methods prune all PW-conv layers of DSConv, inadvertently leading to prune DW-conv layers as well. However, the parameters of DW-conv are only 1.34% of those in PW-conv [41], so each weight element is more sensitive to accuracy, thus for DW-Conv, rather than channel pruning, a more fine-grained pruning is necessary. Therefore, our DEPrune does not directly prune all PW-conv layers. DEPrune applies fine-grained pruning directly to DW-conv and does not prune PW-conv1 directly. We only apply filter pruning to PW-conv2. Pruning only PW-conv2, removes only the parameters in the subsequent DSConv's PW-conv1, which is less sensitive to accuracy drop. Thus, DEPrune effectively prune all the layers of DSConv with high $PR$ and representation power.

## 5 Enhance DEPrune

We propose the following two techniques to enhance DEPrune performance: BWT and HSR.

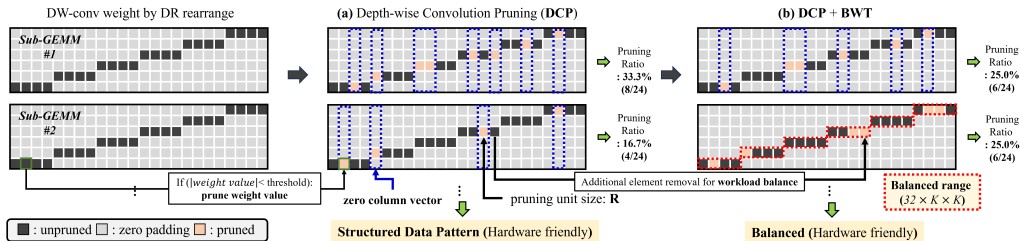

Figure 5: Overview of DCP and Balanced Workload Tuning (BWT). (a) DCP is an element-wise pruning method that creates a structured data pattern. (b) BWT equalizes the $PR$ of all sub-GEMMs. The balanced range of BWT is $32 \times k_h \times k_w$.

## 5.1 DEPrune-B

**(a) Motivation: Imbalance overhead problem of DCP**  GPUs allocate operations of a certain size to streaming multiprocessors (SMs) for massively parallel processing. Therefore, DW-conv's multiple sub-GEMMs are also assigned to SMs, respectively. However, when applying DCP on DW-conv, the pruning ratio ($PR$) of sub-GEMMs may differ, given the varying importance of weights between sub-GEMMs. In that case, the execution time varies for each sub-GEMM due to the difference in $PR$. This results in a workload imbalance problem in that the other SMs of the GPU have to wait until the SM with the lowest $PR$ finishes. The acceleration effect of DCP is then determined by the minimum sub-GEMM $PR$, not by the layer target $PR$. Referring to Fig. 3-(c), the difference between the minimum sub-GEMM $PR$ and the layer target $PR$ is compared for each layer of EfficientNet-B0. In DW Layer 13, when the layer target $PR$ is $60\%$, the minimum sub-GEMM PR is $50.4\%$, which varies up to $9.6\%$, which indicates that it decelerates execution by the amount specified.

**(b) Method: Balanced Workload Tuning (BWT)**  To address the workload imbalance issue of DCP, we propose a DW-conv-specific Workload Balanced Technique that takes into account the operation structure of DW-conv (Fig. 5). DW-conv is a dense matrix where non-zero values are arranged diagonally due to DR, while the remainder consists of zero values. We group all non-zero values within sub-GEMM, which we call a balanced range as illustrated in Fig. 5-(b). Within each balanced range, we rank weight elements with redundancy and systematically prune the lower-ranked elements until the target $PR$ is reached. As every sub-GEMM achieves the same target $PR$ like Fig. 5-(b), this resolves the workload imbalance issue associated with DCP. Since DCP is fine-grained pruning (pruning unit size: $\mathbb{R}$), the representation power loss due to additional BWT is almost negligible. A detailed analysis related to this is in Sec. 6.1.

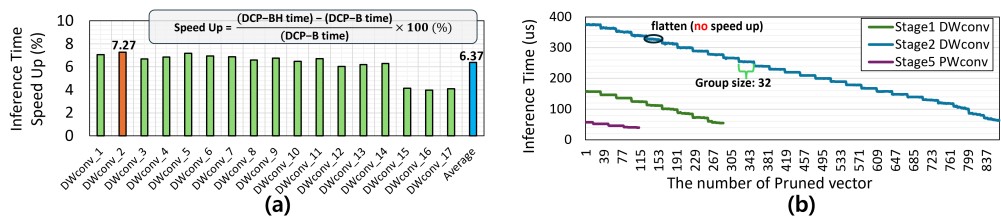

(a)                                      (b)

Figure 6: (a) Measurement of speed increase by layer due to HSR. The orange bar is the max speedup layer. DW-conv $PR$ is 71%. (b) Measurement of DW-conv inference time of EfficientNet-B0 on ImageNet dataset. Inference time decreases with additional pruning of 32 or more vectors. GPU tile size is 32 [14].

## 5.2 DEPrune-BH

**(a) Motivation: Unaligned problem**    As shown in Fig. 7-(a), to maximize parallelism, GPUs divide GEMM operations into small tiles. In general, the size of the tile depends on the hardware specification of GPUs, but it is usually a multiple of 32 [14]. However, if the width of the unpruned weight matrix in Fig. 7-(a) is not a multiple of 32, some parts of the weight tiles are empty. This can cause an unaligned memory access problem on GPUs [11, 13]. In Fig. 6-(b), the inference time does not decrease linearly with an increase in the size of the pruned vector. Whenever the number of pruned vectors increases by 32, the inference time decreases significantly like a step function graph. In DW-conv of Stage 2, the inference time decreases by 7% for each removal of only one tile. Thus, by removing a few additional weight vectors for aligned memory access, we can reduce the inference time by 7% if we align the number of pruned vectors with a multiple of 32 (Fig. 7-(b)).

**(b) Method: Hardware-aware Sparsity Recalibration (HSR)**    We propose Hardware-aware Sparsity Recalibration (HSR) to solve the unaligned memory access problem and enhance DCP-B. As shown in Fig. 7-(c), DCP-B with HSR operates in the following four steps. The first step, DCP-B is applied to DW-conv. The second step, we measure two essential factors ($\alpha$ and $\epsilon$) within the DCP-B model. **(1)** $\alpha$ **:** We measure the speedup obtained by solving the unaligned problem per layer. **(2)** $\epsilon$ **:** We count the number of unpruned vectors of the unaligned tile matrix for each layer. We refer the result obtained by dividing the two parameters, $\alpha$ and $\epsilon$, for each layer as $\beta$. The $\beta$ refers to the size of speed obtained by removing one overflowed vector. The third step, the $\beta$ values of all layers are ranked by comparing them with each other. The last step, the layer with the $\beta$ value of the top 50% is additionally removed as much as it overflows. The additional removed column vector consists of one non-zero value and zero-padded elements. Thus, there is no significant side effect on the representation power. On the other hand, the layer with the $\beta$ value of the bottom 50% additionally recovers as much as it is unaligned. The reason why the criteria for recovery and removal of HSR are set to 50% is to maintain the total target $PR$.

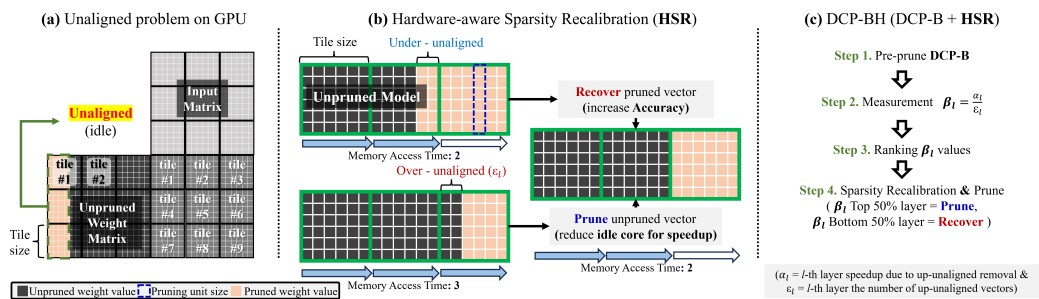

Figure 7: (a) Problem of unaligned pruning ratio on GPU. (b) Concept of Hardware-aware Sparsity Recalibration (HSR). (c) Process of DCP-BH (DCP-B + HSR).

| Model | Method | Pruning Ratio | | | Top-1 Accuracy (%) | | | Speed$^\dagger$ |
|---|---|---|---|---|---|---|---|---|
| | | DW-conv | Real DW | PW-conv | Baseline | Pruned | Diff. | Up |
| MobileNet-V2 | DEPrune | 78% | **71%** | 50% | 71.92 | 71.52 | -0.40 | 2.75× |
| | **DEPrune-B** | 78% | **78%** | 50% | 71.92 | 71.51 | **-0.41** | **3.35×** |
| MobileNet-V3-Small | DEPrune | 82% | **74%** | 50% | 67.67 | 67.26 | -0.41 | 3.88× |
| | **DEPrune-B** | 82% | **82%** | 50% | 67.67 | 67.13 | **-0.54** | **5.09×** |
| EfficientNet-B0 | DEPrune | 85% | **78%** | 40% | 77.69 | 77.01 | -0.68 | 4.51× |
| | **DEPrune-B** | 85% | **85%** | 40% | 77.69 | 77.00 | **-0.69** | **5.79×** |

Table 2: Comparison between DEPrune and DEPrune-B (DEPrune + BWT) on ImageNet dataset. This symbol ($\dagger$) means 'DW-conv inference time speedup than unpruned DW-conv'. 'Real DW' denotes the minimum pruning ratio among the sub-GEMMs of DW-conv. 'Diff.' denotes the difference in Top-1 accuracy between the baseline and pruned models.

| Model | Method | Pruning Ratio | | | Top-1 Accuracy (%) | | | Speed$^\dagger$ |
|---|---|---|---|---|---|---|---|---|
| | | DW-conv | DW-Pat. | PW-conv | Baseline | Pruned | Diff. | Up |
| MobileNet-V2 | DEPrune-B | 77.8% | - | 50.0% | 71.92 | 71.51 | -0.41 | 3.35× |
| | **DEPrune-BH** | 77.9% | **9u8o** | 50.1% | 71.92 | 71.51 | **-0.41** | **3.52×** |
| MobileNet-V3-Small | DEPrune-B | 81.9% | - | 60.2% | 67.67 | 67.17 | -0.50 | 5.09× |
| | **DEPrune-BH** | 82.1% | **6u5o** | 60.0% | 67.67 | 67.18 | **-0.49** | **5.29×** |
| EfficientNet-B0 | DEPrune-B | 84.8% | - | 51.9% | 77.69 | 77.00 | -0.69 | 5.79× |
| | **DEPrune-BH** | 84.7% | **8u7o** | 52.0% | 77.69 | 76.84 | **-0.85** | **6.15×** |

Table 3: Comparison between DEPrune-B and DEPrune-BH (DEPrune-B + DW-conv HSR) on ImageNet dataset. This symbol ($\dagger$) means 'DW-conv inference time speedup than unpruned DW-conv.' 'DW-Pat.' denotes the HSR pattern for DW-conv layers. '**u**' and '**o**' denotes under-aligned and over-aligned layers, respectively. 'Diff.' denotes the difference in Top-1 accuracy between the baseline and pruned models.

# 6 Experiments

We assess the effectiveness of DEPrune using ImageNet [8] and CIFAR-10 [25]. For the validation of image classification, we assess our method with CNN models using DSConv: MobileNet-V2 [43], EfficientNet-B0 [45], and MobileNet-V3 [22].

**Experiment setting on ImageNet**  We utilize pre-trained CNN models sourced from the Pytorch framework [39]. We perform fine-tuning with only 65 epochs after conducting pruning methods. We set a batch size of 256. We use SGD optimizer with the weight decay, $1 \times 10^{-4}$, and the momentum as 0.9 for fine-tuning. The initial learning rate is set to 0.001 and divided by 10 every 30 epoch. All data are augmented with random cropping and horizontal flipping. We evaluate DEPrune using NVIDIA RTX 2080 Ti GPUs [1]. We measured the inference time using NVIDIA CUTLASS [24]. We set the batch size to 32 to measure inference time.

## 6.1 Effect of BWT (DEPrune vs. DEPrune-B)

We analyze the changes in accuracy and speedup resulting from applying the BWT to DEPrune (Table 2). DEPrune has varying pruning ratios among sub-GEMMs, causing the overall speed to be dictated by the sub-GEMM with the smallest pruning ratio. In MobileNet-V2, the smallest sub-GEMM pruning ratio of DEPrune is 71%, as described in the Table 2. Therefore, DEPrune-B in MobileNet-V2 is 21.8% (2.75× → 3.35×) faster in inference time than DEPrune. In MobileNet-V3-Small, DEPrune-B achieves a 31.2% (3.88× → 5.09×) improvement in inference time over DEPrune due to BWT. Since the balanced range of DEPrune-B is significantly large at $32 \times k_h \times k_w$, DEPrune-B has an accuracy drop of within 0.1% than DEPrune across representative models.

## 6.2 Effect of HSR (DEPrune-B vs. DEPrune-BH)

We analyze the changes in accuracy and speedup resulting from the application of the HSR technique to DEPrune-B (Table 3). Since GPUs process operations and memory access in tile units, the actual speed of the GPU does not decrease linearly with the pruning ratio but rather decreases in a step-wise manner, as shown in Fig. 6-(b). By adjusting the pruning ratio to fit the tile size, the DW-conv layer can

| Method | Pruning Ratio | | Pruned FLOPs | Top-1 Accuracy | | | Speed Up | | Time |
|---|---|---|---|---|---|---|---|---|---|
| | DW-conv | PW-conv | | Baseline | Pruned | Diff. | DW-conv | Total | (us) |
| MobileNet-V2* | - | - | - | 71.9% | - | - | 1.00× | 1.00× | 2306 |
| CafeNet-R [44] | 37.1% | 37.1% | | 73.7% | 68.2% | -5.5% | 1.44× | 1.46× | 1581 |
| AMC [19] | - | - | 30.0% | 71.8% | 70.8% | -1.0% | - | - | - |
| CC [29] | - | - | 28.3% | 71.9% | 70.9% | -1.0% | - | - | - |
| MetaPruning [32] | - | - | 30.7% | 72.0% | 71.2% | -0.8% | - | - | - |
| Random-Pruning [28] | - | - | 29.1% | 71.9% | 70.9% | -1.0% | - | - | - |
| ATO [50] | - | - | 30.1% | 71.9% | 72.0% | +0.1% | - | - | - |
| RLAL [12] | - | - | 29.4% | 71.8% | 71.3% | -0.5% | - | - | - |
| GFS [52] | 42.8% | 42.8% | - | 72.0% | 68.8% | -3.2% | 1.58× | 1.60× | 1448 |
| GFS [52] | 37.1% | 37.1% | - | 72.0% | 69.7% | -2.3% | 1.44× | 1.46× | 1581 |
| CafeNet-R [44] | 22.8% | 22.8% | - | 73.7% | 71.9% | -1.8% | 1.22× | 1.23× | 1871 |
| CafeNet-E [44] | 14.2% | 14.2% | - | 73.7% | 72.4% | -1.3% | 1.15× | 1.16× | 1992 |
| AMC [19] | 17.1% | 17.1% | - | 72.0% | 70.8% | -1.2% | 1.17× | 1.20× | 1971 |
| GFS [52] | 22.8% | 22.8% | - | 72.0% | 71.2% | -0.8% | 1.22× | 1.23× | 1871 |
| CafeNet-R [44] | 14.2% | 14.2% | - | 73.7% | 73.3% | -0.4% | 1.15× | 1.16× | 1992 |
| **DEPrune-BH [ours]** | **77.9%** | **52.7%** | 56.1% | 71.9% | 71.6% | **-0.3%** | **3.52×** | **2.48×** | 930 |
| **DEPrune-BH [ours]** | **75.1%** | **64.8%** | 66.2% | 71.9% | 71.0% | **-0.9%** | **3.11×** | **2.70×** | 853 |
| EfficientNet-B0* | - | - | - | 77.6% | - | - | 1.00× | 1.00× | 6650 |
| CafeNet-R [44] | 30.2% | 30.2% | - | 76.4% | 74.5% | -1.9% | 1.41× | 1.37× | 4848 |
| CafeNet-E [44] | 26.4% | 26.4% | - | 76.4% | 74.6% | -1.8% | 1.34× | 1.30× | 5085 |
| **DEPrune-BH [ours]** | **84.7%** | **62.0%** | - | 77.6% | 76.8% | **-0.8%** | **6.15×** | **3.74×** | 1775 |
| MobileNet-V3-Small* | - | - | - | 67.7% | - | - | 1.00× | 1.00× | 1857 |
| GFS [52] | 20.0% | 20.0% | - | 67.5% | 65.8% | -1.7% | 1.24× | 1.23× | 1499 |
| **DEPrune-BH [ours]** | **82.1%** | **70.0%** | - | 67.7% | 67.1% | **-0.6%** | **5.29×** | **4.12×** | 450 |
| MobileNet-V3-Large* | - | - | - | 74.0% | - | - | 1.00× | 1.00× | 4892 |
| FPGM [20] | 33.0% | 33.0% | - | 74.0% | 73.1% | -0.9% | 1.48× | 1.47× | 3945 |
| **DEPrune-BH [ours]** | **77.0%** | **43.0%** | - | 74.0% | 73.7% | **-0.3%** | **4.13×** | **2.83×** | 1187 |

Table 4: Comparison of inference time ($us$) with DEPrune-BH and the latest structured pruning on ImageNet dataset. 'Diff.' denotes the difference in Top-1 accuracy between the baseline and pruned models. DEPrune-BH applies filter pruning using $\ell_2$-norm to PW-conv [26]. This symbol ($\star$) means 'baseline model'.

achieve an average inference time speedup of 6.37%, as illustrated in Fig. 6-(a). According to Table 3, applying HSR to DEPrune-B shows almost no difference in accuracy compared to not applying HSR within 0.16%. Specifically, the accuracy difference is only 0.01% on MobileNet-V3-Small. For DEPrune-BH, all models have nearly identical numbers of over-aligned and under-aligned layers. DEPrune-BH achieves 6.2% ($5.79\times \rightarrow 6.15\times$) inference time speedup compared to DEPrune-B on EfficientNet-B0. Additionally, HSR can be applied to PW-conv layers as well.

## 6.3 Comparison with Structured Pruning

In Table 4, we conduct experiments comparing DEPrune with the latest structured pruning methods across four models. On MobileNet-V2, our DEPrune-BH reduces approximately 26.7% more FLOPs compared to RLAL, while exhibiting a 0.2% smaller accuracy drop. GFS removes up to 42.8% of DSConv parameters, resulting in an accuracy drop exceeding 3%. In contrast, DEPrune-BH eliminates 75.1% and 64.8% of parameters in DW-conv and PW-conv, respectively, with an accuracy drop within 1%. On EfficientNet-B0, while other methods prune around 30% of DW-conv, our method prunes 84.7% with only a 0.8% accuracy drop. On MobileNet-V3-Small and MobileNet-V3-Large, DEPrune-BH achieves inference times 3.33 times and 3.32 times faster than GFS and FPGM, with accuracy drops of 1.1% and 0.6% less, respectively.

## 6.4 Discussion: Various Pruning on PW-conv

We apply four structured pruning techniques to PW-conv layers to measure the changes in accuracy (See Table 5). When applying $\ell_1$-norm pruning and $\ell_2$-norm pruning to PW-conv layers, the accuracy difference is within 0.06% for all models except EfficientNet-B0. According to the FP paper [27], there is minimal difference between $\ell_1$-norm and $\ell_2$-norm pruning, and this similarity is also observed in the case of DEPrune-BH. Conversely, on EfficientNet-B0, FPGM [20] which uses geometric median achieves 0.33%, and 0.09% higher accuracy compared to $\ell_1$-norm and $\ell_2$-norm pruning, respectively. BCBP [37] is a block-wise pruning method that can be applied PW-conv. However, applying BCBP to PW-conv following DW-conv eliminates some parameters of DW-conv. Therefore,

| Model | Method | | Pruning Ratio | | Top-1 Accuracy (%) | | |
|-------|--------|--------|--------|--------|--------|--------|--------|
| | DW-conv | PW-conv | DW-conv | PW-conv | Baseline | Pruned | Diff. |
| MobileNet V2 | DEPrune-BH [**ours**] | **FP** ($\ell_1$-norm) [27] | 78% | 40% | 71.92 | **71.59** | **-0.33** |
| | | FP ($\ell_2$-norm) [27] | 78% | 40% | 71.92 | 71.54 | -0.38 |
| | | FPGM [20] | 78% | 40% | 71.92 | 71.42 | -0.50 |
| | | BCBP [37] | 78% | 40% | 71.92 | 71.23 | -0.69 |
| MobileNet V3 Small | DEPrune-BH [**ours**] | FP ($\ell_1$-norm) [27] | 82% | 50% | 67.67 | 67.11 | -0.56 |
| | | FP ($\ell_2$-norm) [27] | 82% | 50% | 67.67 | 67.17 | -0.50 |
| | | **FPGM** [20] | 82% | 50% | 67.67 | **67.18** | **-0.49** |
| | | BCBP [37] | 82% | 50% | 67.67 | 66.09 | -1.58 |
| EfficientNet B0 | DEPrune-BH [**ours**] | FP ($\ell_1$-norm) [27] | 85% | 52% | 77.69 | 76.60 | -1.09 |
| | | FP ($\ell_2$-norm) [27] | 85% | 52% | 77.69 | 76.83 | -0.85 |
| | | **FPGM** [20] | 85% | 52% | 77.69 | **76.93** | **-0.76** |
| | | BCBP [37] | 85% | 52% | 77.69 | 75.91 | -1.78 |

Table 5: Comparison with various pruning methods [27, 20, 37] applied to PW-conv on ImageNet dataset. 'Diff.' denotes the difference in Top-1 accuracy between the baseline and pruned models.

when applying BCBP to PW-conv the accuracy drops on all models described in Table 5 compared to FP and FPGM.

# 7 Conclusion

In this work, we propose a new Depth-wise Separable Convolution Pruning (DEPrune) method tailored for DW-conv to reduce DSConv inference time and fully leverage GPU features. Extensive experimental results on the ImageNet dataset demonstrate that DEPrune effectively preserves representation power, even with higher $PR$ than structured pruning, achieving a regular sparsity. Moreover, two techniques, BWT and HSR, further enhance DEPrune's capabilities. With these combined features, DEPrune-BH achieves substantial GPU speed gain of up to $4.1\times$ on MobileNet-V3-Small.

# 8 Acknowledgements

This work was partly supported by Institute of Information & communications Technology Planning & Evaluation (IITP) grant funded by the Korea government (MSIT) (No.RS-2024-00402898, Simulation-based High-speed/High-Accuracy Data Center Workload/System Analysis Platform), (RS-2021-II212068, Artificial Intelligence Innovation Hub), and the National Research Foundation of Korea (NRF) grant funded by the Korea government (MSIT) (RS-2024-00357037).

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

# A  Appendix

## A.1  Limitation of Balanced Pruning

A number of pruning studies [51, 37, 58, 7, 31] have been proposed to balanced pruning. In a high pruning ratio, the balanced pruned model has negligible accuracy drop. However, achieving a balanced pruned model's speedup is difficult without support for specific hardware architecture or kernel. Therefore, NVIDIA has changed its hardware structure for balanced pruning [7].

## A.2  Why Balanced Pruning Cannot Be Applied to DW-conv?

Balanced pruning [51] is a pruning method that accelerates performance by considering workload balance. This approach divides the weight matrix into consistent vector units and assigns an identical $PR$ to each vector. The designated vector range is referred to as a 'balanced range'. Elements within the vector are ranked based on their redundancy, and those with the lower ranks are pruned sequentially until the target $PR$ is achieved. However, directly applying traditional balanced pruning to DW-conv is not suitable. This is because the balanced pruning method doesn't take into account the structure of DW-conv. Since non-zero value weights in DW-conv with DR are arranged diagonally when setting a balanced range, the majority of the values within that range end up being zeros.

## A.3  Limitation

The limitation of DEPrune is that it is specialized for DW-conv. Specifically, DEPrune is difficult to apply to the Vision Transformer series. Although some models in the Vision Transformer [4] use DW-conv, most of the computations in Vision Transformers are performed using self-attention and feed-forward network layers. However, our proposed HSR technique appears to be applicable to these layers.

## A.4  Experiments on CIFAR-10 Dataset

We experiment the effectiveness of our proposed method using the CIFAR-10 dataset [25]. For the validation of image classification, we experiment our method with CNN models: MobileNet-V2 [43] and EfficientNet-B0 [45] We perform fine-tuning with only 100 epochs after processing pruning methods on CIFAR-10.

## A.5  Limitation of Channel Pruning on DW-conv

When channel pruning is applied to depth-wise convolution, the pruned model has a structured data pattern. However, the pruning unit size of channel pruning is $k_h \times k_w$. Since channel pruning is $9$ $(3^2)$ or $25$ $(5^2)$ times larger than DEPrune with a pruning unit size of 1, thereby channel pruning reduces more representation power than DEPrune. In Mobilenet-V2 on CIFAR-10, when $PR$ is $50\%$, the difference in accuracy between DEPrune and channel pruning is $0.23\%$ (See Table 6). Even when $PR$ is $70\%$, the difference of accuracy is $0.35\%$. Therefore, our proposed DEPrune, which has not only a structured data pattern but also representation power advantage, is appropriate for DW-conv.

| MobileNet-V2 on CIFAR-10 | | | |
|---|---|---|---|
| Pruning Ratio | | Accuracy | |
| DW-conv | PW-conv | Channel Pruning | DEPrune |
| 50% | 30% | 93.07% | 93.30% |
| 60% | 30% | 92.95% | 92.99% |
| 70% | 30% | 92.45% | 92.80% |

Table 6: Comparison of accuracy between DEPrune and Channel Pruning with MobileNet-V2 on CIFAR-10 dataset.

## A.6  Comparison between DCP and Filter Pruning on PW-conv

When filter pruning is applied to PW-conv, the PW-conv pruned model has a structured data pattern. If filter pruning is performed on PW-conv, the channel of DW-conv that follows is also removed. Even

the channel of PW-conv behind DW-conv is removed as well. Therefore, the pruning unit size of filter pruning is not simply a filter of the corresponding PW-conv. The pruning unit size includes also the parameters of following layers. The larger the pruning unit size, the greater the probability that a core parameter is included in the removing group, which influences the representation power. On the other hand, our proposed DCP does not remove the parameters of other layers. Therefore, when the $PR$ is 70%, our DCP is 0.62% higher in representation power than filter pruning of PW-conv (See Fig. 8).

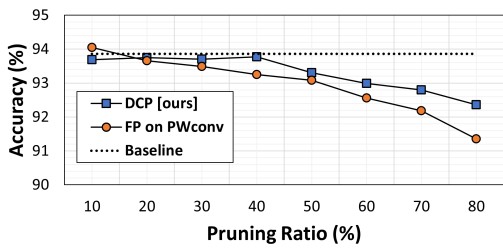

Figure 8: Comparison of accuracy (%) with DCP and filter pruning (FP) on PW-conv of MobileNet-V2 on CIFAR-10.

## A.7    Effect of Balanced Range

Research on n:m sparsity is currently very active topic in the field of pruning. However, this sparsity approach has two major limitations: a lack of flexibility and the requirement for specialized hardware. First, it lacks flexibility because it is fixed at a 50% pruning ratio, specifically 2:4 pruning [37]. As seen in Table 7, we conducted comparative experiments between NVIDIA's n:m sparsity and DEPrune on MobileNet-V2 using CIFAR-10. At the same pruning ratio of 50%, DEPrune-B achieves 0.31% higher accuracy than n:m sparsity. This is because DEPrune-B achieves a 50% pruning ratio within total parameters, whereas n:m sparsity achieves a 50% pruning ratio within a parameter size of 4 [23]. Secondly, in n:m sparsity, achieving optimal performance requires specialized hardware (NVIDIA A100) that can quickly handle index processing [37]. In contrast, our approach requires only a customized GPU kernel for execution.

| MobileNet-V2 on CIFAR-10 | | | |
|---|---|---|---|
| Method | Pruning Ratio | Accuracy | Diff. |
| MobileNet-V2 | - | 93.86% | - |
| NVIDIA n:m sparsity | 50% | 92.99% | -0.87% |
| DEPrune-B | 50% | 93.30% | -0.56% |

Table 7: Comparison of accuracy (%) with DEPrune-B and NVIDIA n:m pruning on CIFAR-10 dataset. 'Diff.' denotes the difference in accuracy between the baseline and pruned model. NVIDIA n:m pruning's n and m size are 2 and 4. DEPrune-B applies filter pruning using $\ell_2$-norm to PW-conv.

## A.8    Effect of Balanced Workload Tuning (BWT)

We propose Balanced Workload Tuning (BWT) to solve the workload imbalance problem on DCP (Sec. 5.1). However, the BWT method may slightly reduce the representation power. As shown in Table 8, we compare DCP-B (DCP + BWT) with DCP on EfficientNet-B0 of CIFAR-10. When $PR$ is 50%, the difference between the DCP-B and DCP accuracy is 0.17%. Therefore, there is little representation power loss due to BWT.

## A.9    Peak Memory Usage

According to paper [41], the extra overhead in total memory consumption due to zero-padding is approximately 0.3%. To assess the impact of DEPrune-BH, we measured and presented the peak memory usage of MobileNet-V2 before and after applying DEPrune-BH with a 50% pruning ratio, as

| EfficientNet-B0 on CIFAR-10 | | | |
|---|---|---|---|
| Pruning Ratio | | Accuracy | |
| DW-conv | PW-conv | DCP | DCP-B |
| 10% | 10% | 91.25% | 91.49% |
| 20% | 10% | 91.18% | 91.31% |
| 50% | 10% | 91.16% | 91.33% |

Table 8: Comparison between DCP and DCP-B of EfficientNet-B0 on CIFAR-10 dataset.

shown in Table 9. Before applying DEPrune-BH, the peak memory usage is 7.22 MB, whereas after application, it decreases to 3.63 MB, representing a reduction of approximately 49.8%.

| Peak Memory Usage (MobileNet-V2 on ImageNet) | | | |
|---|---|---|---|
| Pruning Method : DEPrune-BH | | | |
| Pruning Ratio | Pre-pruning | After-pruning | GAP |
| 50% | 7.22 MB | 3.63 MB | 3.59 MB |

Table 9: Analysis of Peak Memory Usage (MB) with DEPrune-BH on ImageNet dataset. 'GAP' means the after-pruning peak memory usage difference rate compared to pre-pruning. DEPrune-BH applies filter pruning using $\ell_2$-norm to PW-conv.

