# OpenReview forum: "DEPrune: Depth-wise Separable Convolution Pruning for Maximizing GPU Parallelism"
_NeurIPS.cc/2024/Conference — NeurIPS 2024 poster_

### Official Review · Reviewer_VAUi · 2024-07-12

**Soundness:** 2
**Presentation:** 3
**Contribution:** 2
**Rating:** 6
**Confidence:** 3

**Summary:**

This paper proposed a pruning method target at the previously neglected Depth-wise Convolution layers which are widely used in vision models. This work is based on the Diagonal-wise Refactorization (DR) computation strategy used in GPU, and pruning one weight point in depth-wise convolution means convert the corresponding column into zero vector. To accelerate the inference of pruned models, the authors proposed to enhance the GPU performance with two hardware friendly methods: BWT and HSR, which achieves the load balance and align the memory access with GPU tile size. The evaluation result shows the proposed framework brings significant speedup in MobileNet and EfficientNet.

**Strengths:**

1. The pruning method is based on the DR computation strategy, which provides the opportunity to convert the point pruning in depth-wise convolution layers into masking the corresponding columns as zeros.
2. To accelerate the inference processing, the proposed BWT and HSR method achieves loading balance among GPU kernels and align the computation with tile size, speeding up the GPU processing.
3. The evaluation result breakdowns the pruning method, BWT and HSR, demonstrating the contribution of each part.

**Weaknesses:**

1. The overhead of pruning method is not analyzed. How much time does it need to decide pruning points, loading balance and the parameters in HSR?
2. Comparing with the gain in pruning ratio, the accuracy drop seems not negligible. Can we modify the algorithm to set the HSR constraints to align the memory access and computation with tile size in pruning phase, without need of recalibration? Does it help with accuracy?
3. Can you provide the evaluation on edge GPU?

**Questions:**

See weakness.

**Limitations:**

See weakness.

---

> ### Author Rebuttal · Authors · 2024-08-07
>
> Thank you for your support of our paper and the valuable feedback. We respond to your reviews as follows.[m-#] means manuscript's reference.
>
> **Weakness 1) [The overhead of pruning method is not analyzed. How much time does it need to decide pruning points, loading balance and the parameters in HSR?]**
>
> We would like to clarify few things that might not have been clearly expressed in the paper. The pruning, load balancing, and HSR mentioned are all required only during the pruning process (offline). Pre-processing (Pruning and fine-tuning) is conducted offline, while model inference is performed online. The processes you mentioned are overheads in the offline phase and do not affect model inference time. In Author-Rebuttal-Table.4, still this offline overhead accounts for about 0.6\% of the total pre-processing time in MobileNet-v3-small on ImageNet. The proportion of this overhead becomes even smaller if the fine-tuning epoch increases. We will provide detailed figures based on MobileNet-v3-small to Author-Rebuttal-Table.4.
>
>
> **Weakness 2) [Comparing with the gain in pruning ratio, the accuracy drop seems not negligible. Can we modify the algorithm to set the HSR constraints to align the memory access and computation with tile size in pruning phase, without need of recalibration? Does it help with accuracy?]**
>
> Yes, the algorithm can be modified as you suggested. However, the absence of recalibration does not significantly affect accuracy.  Without the recalibration process, users must set the pruning ratio to match the recalibration results by considering the GPU tile size and each layer's size.  Recalibration is an algorithm that automates this process. The main reason there is no significant change in accuracy is as follows: In Figure 7-(C), you can see that DEPrune-BH performs fine-tuning after step 4, which is the entire pruning setting process. Therefore, whether recalibration is performed or users manually set the pruning ratio by considering the tile size, both are processes before fine-tuning, resulting in almost no change in accuracy.
>
>
>
> **Weakness 3) [Can you provide the evaluation on edge GPU?]**
>
> Certainly. We conducted additional experiments on an edge GPU and measured the inference time before and after pruning for MobileNet-v2. The edge GPU that we used for the experiment is NVIDIA Jetson Orin Nano 8GB. In Author-Rebuttal-Table.5, the DEPrune-BH pruned model inference time shows a 2.48 times speedup in inference time than baseline (unpruned model) on edge GPU.  Compared to GFS [m-50] pruned model, DEPrune-BH pruned model is approximately 1.62 times faster. From an accuracy perspective, there is no change because the weights are the same as those on the laptop GPU.

---

> > ### Comment · Reviewer_VAUi · 2024-08-12
> >
> > Thank you for your clarification, which addresses my concerns. I would like to improve the score to 6.

---

### Official Review · Reviewer_bd55 · 2024-07-12

**Soundness:** 3
**Presentation:** 2
**Contribution:** 3
**Rating:** 6
**Confidence:** 4

**Summary:**

This paper presents the Depth-wise Separable Convolution Pruning (DEPrune). DEPrune prunes point-wise convolution and depth-wise convolution (DW-conv). And it is optimized by considering and analyzing the computation of Depth-wise Separable Convolution on GPU. Experimental results validate the speedup of DSPrune.

**Strengths:**

1.	The proposed method supports both point-wise convolution and depth-wise convolution pruning.
2.	The performance of the proposed method seems significant.

**Weaknesses:**

1.	Some references are incomplete, for example, [23][39].
2.	The proposed scheme is only evaluated on MobileNet-V2, MobileNet-V3, and EfficientNet-B0. How does this method work for other NN models?

**Questions:**

1.	What does “X” and “O” mean in Figure 1?
2.	The lower parts of Figure 1 seem misleading, which do not correspond to the DW-conv example in the upper half.

**Limitations:**

Yes.

---

> ### Author Rebuttal · Authors · 2024-08-07
>
> Thank you for the time you have taken to review our work and for the constructive feedback. If the reviewer allows us to revise this paper, we will try our best to improve the quality of this paper.
>
>
> **Weakness 1) [Some references are incomplete, for example, [23][39].]**
>
> We greatly appreciate your guidance on this issue. We will revise and incorporate your suggestions as follows. Additionally, we will double-check if other references are incomplete and address them accordingly.
>
> "[23] Krizhevsky, G. Hinton, et al. Learning Multiple Layers of Features from Tiny Images. Technical report, University of Toronto, 2009."
>
> "[39] P. K. Rao and S. Chatterjee. Weight Pruning-UNet: Weight Pruning UNet with Depth-wise Separable Convolutions for Semantic Segmentation of Kidney Tumors. Research Square 2021"
>
>
> **Weakness 2) [The proposed scheme is only evaluated on MobileNet-V2, MobileNet-V3, and EfficientNet-B0. How does this method work for other NN models?]**
>
> As the reviewer's suggestion, we conducted additional experiments with the ConvNeXt-Base model [1].  In Author-Rebuttal-Table.3, DEPrune-BH achieves a 0.46\% higher Top-1 accuracy compared to FP even with an additional 20\% pruning in DW-conv. DEPrune-BH is up to 3.3 times faster than the ConvNeXt-Base while achieving a Top-1 accuracy drop within 1.5\%.
>
> [1] A ConvNet for the 2020s, CVPR 2022
>
>
> **Question 1) [What does “X” and “O” mean in Figure 1?]**
>
> We agree with the reviewer’s comment that the caption in Figure 1 is not clearly presented. The X and O symbols indicate the absence and presence of corresponding characteristics for each method.
> In the case of (a) Structured Pruning, this approach forms well-structured data patterns that can benefit performance when executed on GPUs. However, it is not a fine-grained method, as the pruning unit size can extend up to a 2D matrix, which is significantly larger than the pruning unit in DEPrune, the method we propose. Therefore, in the case of (b) DEPrune, it is marked as O for both fine-grained and structured characteristics. We will include these details in the final version of the paper’s caption to prevent any misunderstandings for readers.
>
>
> **Question 2) [The lower parts of Figure 1 seem misleading, which do not correspond to the DW-conv example in the upper half.]**
>
> We appreciate the reviewer’s valuable comment that points out the statements that could possibly mislead the readers. The top illustration in Figure 1 depicts a typical depth-wise convolution. The bottom illustration shows the depth-wise convolution rearranged to be more favorable for GPU processing. On the left is an image with (a) structured pruning applied, and on the right is after applying (b) DEPrune. Thus, the relationship between the top and bottom illustrations indicates before and after the Diagonalwise Refactorization is applied, and the figure shows the arrangement of the weight. We will enhance the caption with these details to ensure there is no misunderstanding for readers.

---

> > ### Comment · Reviewer_bd55 · 2024-08-12
> >
> > Thanks the authors’ reply. My concerns are addressed.

---

### Official Review · Reviewer_CSWm · 2024-07-16

**Soundness:** 3
**Presentation:** 2
**Contribution:** 3
**Rating:** 7
**Confidence:** 4

**Summary:**

This paper presents DEPrune, which prunes depthwise separable convolution models into a structured sparsity pattern that is friendly to depthwise refactorization.

**Strengths:**

1. Much of the motivation for conducting CNN pruning is to pursue the most lightweight — often time edge-deployed — model run under much resource constraint. Depthwise separable convolution (DSConv)-based models have a proven track record in this regard, and it is true that there are very few attempts to investigate how to prune such models. So, the exploration done in this paper is a welcoming addition to the field.

2. The authors clearly demonstrate the further complications caused by filter pruning DSConv models and address them by proposing a structured sparsity pattern that is friendly to depthwise refactorization — a common application for improving the utility of DSConv — along with many other submodules. The DEPruned models show real end-to-end speedup.

3. Tables 3 & 4 show DEPrune delivers better accuracy and runtime results than applying typical structured pruning methods to DSConv models.

**Weaknesses:**

1. Most of the structured pruning methods compared in Table 4 is a bit old. Although I suspect the conclusion won't change much due to the gap present and the fact that methods like CC [1] are often plenty performant even for today, still, a comparison with more modern structured pruning methods should be included.

2. The authors emphasize that "there are no previous pruning techniques that even target depth-wise convolution (DW-conv)" in line 4. What about methods like [2] that address the exact same DSConv?

3. The paper did a good job explaining each of its components in detail but lacks a clear high-level overview of how these components are attached together — some abstract visualization with a detailed walkthrough in the caption, as well as a pseudo-code section, should be added. Similarly, the experiment procedure lacks clarity: only the epoch budget of CIFAR10 experiments is shared, and critical details like the learning rate for finetuning, whether the process is iterative or post-prune, etc, are missing.

4. No throughput or latency (bs = 1) results.

5. Not really a weakness, but DSConv pruning is a relatively niche art with limited familiarity with the pruning community. This paper would benefit from making better contrasts with standard conv pruning to highlight its unique challenges.

6. Is this DSPrune or DEPrune? The proposed method family has so many submodules named with acronyms, and it could really use a richer Table 1 for a more friendly reading experience.

[1] Towards Compact CNNs via Collaborative Compression
[2] Pruning Depthwise Separable Convolutions for Extra Efficiency Gain of Lightweight Models
[3] Revisit Kernel Pruning with Lottery Regulated Grouped Convolutions
[4] Structured Sparsity in the NVIDIA Ampere Architecture and Applications in Search Engines

**Questions:**

1. Why are the efficiency metrics of any methods missing in Table 4?
2. Can the DCP part be applied to general group convolution (where the number of groups != number of channels)?
3. Following Q1, some of the structured pruning methods in the GKP [3] series are known to be able to maintain the output shape of a pruned conv layer by leveraging group convolution. Can this be applied to the PWConv part of the DSConv and therefore avoid the concerns introduced in Section 4.2 and Appendix 6?
4. Is a DEPruned model zero-padded? It looks like the case in Figure 4. If so, what kind of peak memory usage are we seeing pre-and-after pruning?
5. How would this work compare to general structured sparsity approaches with more pruning freedom? e.g., the N:M sparsity [4]?

**Limitations:**

The authors addressed the limitations.

---

> ### Author Rebuttal · Authors · 2024-08-07
>
> We sincerely appreciate the reviewer’s time and effort in providing constructive feedback on this manuscript. We will incorporate all of the suggestions into the final version of the paper. [m-\#] means manuscript's reference.
>
>
> **Weakness 1)**
>
> As the reviewer pointed out, it is important to compare our proposed scheme with recent prior work. In Table 4, we have compared our scheme with two of the most recent research papers [m-11 and m-48], both presented in 2024. If you have any recommendations for recent structured pruning methods, we would be eager to compare them with our DEPrune if circumstances allow.
>
>
> **Weakness 2)**
>
> We appreciate your comment. The study you mentioned [2 and m-44] indeed applies an existing gradual pruning method to DSConv, so it is not accurate to state that there are no previous pruning techniques that target DW-Conv. We will revise this statement in the paper if given the opportunity to make revisions: "there are hardly any previous pruning techniques that target depth-wise convolution (DW-conv)" in line 4.
> We also want to mention that the pruning method proposed in [2 and m-44] is focused on applying gradual pruning [m-55] to DSConv layers, which differs from our approach.
> The research [2] focuses on removing filters to preserve structural pattern on DW-conv. Whereas our research considers GPU kernel and maintains structural pattern while more fine-grained pruning on DW-conv, to reduce accuracy loss.
>
>
> **Weakness 3)**
>
> Thank you for your valuable advice. Based on your feedback, we will make the following improvements:
> 1. We will rewrite the captions to provide a more detailed walkthrough, ensuring that readers can easily understand them.
> 2. To enhance reproducibility, we will add a pseudo-code section to the appendix and plan to release our implementation in the future.
> 3. We will include the ImageNet dataset information in the experiment section to clearly outline the experimental procedure:
>
> _[ImageNet experiment setting]
> learning rate : 0.001, Fine-tuning epoch : 65epochs, learning rate epoch : dividing 10 per every 30epochs, Process : iterative pruning, Weight decay : 1e-4, Optimizer : SGD, Momentum : 0.9, Pre-trained model : PyTorch, All data are augmented with random crop and randomly horizontal flip._
>
>
> **Weakness 6)**
>
> Thank you for pointing out our typo. Our proposed scheme is named DEPrune, and the mention of DSPrune is indeed a typo. We will thoroughly review our paper again and correct this and any other typographical errors in the revised version.
> We also think that so many names can be very difficult to understand. We will double-check the Table 1 and writing of the paper so that it can provide simple and clear concept of DEPrune. As you suggested, creating a richer table will likely make it easier for readers to understand the paper.
>
>
> **Question 1)**
>
> We believe there are various efficiency metrics, such as peak memory usage, computation throughput, and computation utilization, that can be measured in our experiments. Among these, we consider peak memory usage to be the most important efficiency metric, thus, we will include it in Table 4 in the revised paper. If the reviewer can suggest other efficiency metrics that should be included, we will measure them and incorporate the results in the revised version.
>
>
> **Question 2)**
>
> Applying DCP to general group convolution can be challenging. When the number of groups and channels differ, the rearrangement method for GPU processing varies, complicating the application of DCP. However, the enhanced method HSR, proposed in our paper, can be effectively applied in these scenarios. We believe that combining conventional structured pruning with our HSR could lead to improved inference time performance on the general group convolution.
>
>
> **Question 3)**
>
> Thank you very much for your insightful feedback. The paper you mentioned, GKP [3], seems an excellent work. GKP divides weights into groups and applies vector-wise pruning to each group, effectively addressing the issue described in Section 4.2. As you pointed out, when GKP is applied, it does not impact the subsequent layer, thereby avoiding the problem discussed in Section 4.2. We appreciate you bringing this important paper to our attention, and we will consider incorporating this research (GKP) with our DEPrune as a potential future work.
>
>
> **Question 4)**
>
> According to Table 4 of paper [m-38], the extra overhead in total memory consumption due to zero-padding is approximately 0.3\%. To assess the impact of DEPrune-BH, we measured and presented the peak memory usage of MobileNet-v2 before and after applying DEPrune-BH with a 50\% pruning ratio, as shown in Author-Rebuttal-Table 1. Before applying DEPrune-BH, the peak memory usage is 7.22 MB, whereas after application, it decreases to 3.63 MB, representing a reduction of approximately 49.8\%.
>
>
> **Question 5)**
>
> Research on n:m sparsity is currently very active in the field of pruning. However, this sparsity approach has two major limitations: a lack of flexibility and the requirement for specialized hardware.
> First, it lacks flexibility because it is fixed at a 50\% pruning ratio, specifically 2:4 pruning [m-35].
> As seen in Author-Rebuttal-Table.2, we conducted comparative experiments between NVIDIA's n:m sparsity and DEPrune on MobileNet-v2 using CIFAR-10.
> At the same pruning ratio of 50\%, DEPrune-B achieves 0.31\% higher accuracy than n:m sparsity.
> This is because DEPrune-B achieves a 50\% pruning ratio within $32 \times k \times k$ parameters, whereas n:m sparsity achieves a 50\% pruning ratio within a parameter size of 4, which is $8 \times k \times k$ times smaller than DEPrune-B, leading to a accuracy drop [m-21].
> Secondly, in n:m sparsity, achieving optimal performance requires specialized hardware (NVIDIA A100) that can quickly handle index processing [m-35]. In contrast, our approach requires only a customized GPU kernel for processing.

---

> > ### Comment · Reviewer_CSWm · 2024-08-11
> > **Thank you for your rebuttal. Can we have the bs=1 latency report (W4)?**
> >
> > I am satisfied with the authors' responses. However, it looks like W4 remains unaddressed. Is it possible to have a latency report with a batch size of 1?
> >
> > My original Q1 has a typo. It should be "Why are the efficiency metrics of some methods missing in Table 4?" I hope the authors can also follow up on this. Thanks.

---

> > > ### Author Response · Authors · 2024-08-13
> > >
> > > We sincerely thank you for the additional feedback and for rephrasing the question about efficiency metrics.
> > >
> > > [m-\#] means manuscript's reference.
> > >
> > > **Weakness 4) [Is it possible to have a latency report with a batch size of 1?]**
> > >
> > > We apologize that we could not respond to your comments on W4 due to the word limit (restricted to within 6,000 characters).
> > > The inference time using a batch size of 1 is provided in Rebuttal-Table 1.
> > > As shown in Rebuttal-Table 1, when the batch size is reduced to 1, DEPrune-BH demonstrates 1.90 and 1.57 times faster performance compared to the unpruned model and CafeNet-R [m-41], respectively.
> > >
> > > We observe reduced pruning's performance improvements when the batch size decreases from 32 to 1.
> > > The primary reason for this degradation is that GPUs rely heavily on a high level of thread-level parallelism (TLP).
> > > As the batch size increases, the number of scheduled threads (corresponding to the size of GEMM operations in NNs) also increases, allowing GPUs to achieve significant performance gains due to their ability to execute many threads simultaneously.
> > > Conversely, when the batch size is reduced, the degrees of TLP decrease (with the size of GEMM operations becoming smaller in NNs), leading to underutilization of GPU computation units and resulting in many execution units sitting idle for longer periods.
> > > Due to various characteristics of the GPU, reducing the batch size does not lead to a linear decrease in latency [1,2].
> > >
> > > Due to the limited time window of the rebuttal, we were only able to measure the performance of EfficientNet-B0 at this time.
> > > In the future, we will share additional data for other models as well.
> > > Thank you for the helpful review.
> > >
> > > [1] "Accuracy-constrained efficiency optimization and GPU profiling of CNN inference for detecting drainage crossing locations." Proceedings of the SC'23 Workshops of The International Conference on High Performance Computing, Network, Storage, and Analysis. 2023.
> > >
> > > [2] "Variable batch size across layers for efficient prediction on CNNs." 2020 IEEE 13th International Conference on Cloud Computing (CLOUD).
> > >
> > >
> > > |     **Method**    | **DW-conv Pruning Ratio** | **PW-conv Pruning Ratio** | **Time (us)** | **Speed Up** |
> > > |:-----------------:|:-------------------------:|:-------------------------:|:-------------:|:------------:|
> > > |  EfficientNet-B0  |             -             |             -             |      1276     |     1.00x    |
> > > |     CafeNet-R [m-41]     |           30.2\%           |           30.2\%           |      1055     |     1.21x    |
> > > |     CafeNet-E [m-41]    |           26.4\%           |           26.4\%           |      1072     |     1.19x    |
> > > |     DEPrune-BH    |           84.7\%           |           62.0\%           |      672     |     1.90x    |
> > >
> > > [Rebuttal-Table.1 : Comparison of inference time with DEPrune-BH and other pruning on ImageNet. We set the batch size to 1.]
> > >
> > > **Question 1) [Why are the efficiency metrics of some methods missing in Table 4?]**
> > >
> > > Thank you for the clarification.
> > > The reason some metrics are not included in Table 4 is that the official experimental data were not disclosed in the corresponding pruning paper.
> > > To ensure a fair comparison, we only included the metrics provided in the official paper.
> > > Since some models only reported pruned FLOPs and did not provide pruned parameters, it is difficult to accurately measure the exact inference time based on that information alone.
> > > For those models that provide pruned parameters, we measured only inference time of the pruning methods by carefully reviewing the available information.

---

> > > > ### Comment · Reviewer_CSWm · 2024-08-13
> > > > **Thank you, bumping it to 7.**
> > > >
> > > > Please just make sure you'd incorpoerate the rebuttal material in your camera-ready paper.

---

> > > > > ### Author Response · Authors · 2024-08-14
> > > > > **Thank you for your valuable feedback.**
> > > > >
> > > > > Thank you for your valuable feedback. We will make sure to incorporate the rebuttal material into the revised manuscript.

---

### Official Review · Reviewer_cCXz · 2024-07-17

**Soundness:** 4
**Presentation:** 3
**Contribution:** 4
**Rating:** 7
**Confidence:** 4

**Summary:**

The paper addresses an important topic of pruning depth-wise separable convolutions (DSConv) called DEPrune. While structural model pruning methods like channel pruning can achieve significant speed-up for regular convolutions, they cannot secure notable speed-up on DSConv layers as they mainly prune the point-wise convolution, which is a small fraction of the compute for DSConvs. Further, naively pruning the depth-wise channels significantly reduces its capacity, leading to severe performance degradation. DEPrune prunes the depth-wise convolutions in a fine-grained manner yet achieves structural sparsity to enable practical speed-up on GPUs. It also introduces two techniques BWT and HSR to further improve the performance.

**Strengths:**

1. The paper is well-written. The motivations behind the design choices are clearly explained.

2. The problem of pruning DSConvs is an important topic as DSConvs are widely adopted for edge applications like MobileNets. Therefore, further increasing their efficiency is of significant interest to the community.

3. Extensive experimental results validate the design choices and show clear speed-up compared to the original models.

**Weaknesses:**

I cannot think of a major weakness in the paper. I hope that the authors provide more background for the readers not directly familiar with the concepts of "alignment in GPUs" or refer to proper references.

**Questions:**

I think the paper is interesting and can be useful in practical applications. Currently, as far as I know, DSConvs are not used in computationally intensive architectures like the U-Net models in diffusion models. Eventually, when more efficient architectures using DSConvs are developed, the proposed DEPrune can further improve the performance. I hope the authors release their implementations so that the community can benefit from them.

**Limitations:**

The limitations are explained in the supplementary.

---

> ### Author Rebuttal · Authors · 2024-08-07
>
> We would like to express our gratitude to the reviewer for their constructive and insightful feedback. Below, we have provided our responses to the comments. If the paper is accepted, we will incorporate these helpful suggestions into the camera-ready version.
>
> **Weakness 1) [I hope that the authors provide more background for the readers not directly familiar with the concepts of "alignment in GPUs" or refer to proper references.]**
>
> We agree with the reviewer’s comment that the concepts of "alignment in GPUs" should be presented in our paper for better understanding. Therefore, if given the opportunity to revise our paper, we will include these details, referencing prior work [1, 2].
>
> [1] Sparse GPU kernels for Deep Learning, SC 2020
>
> [2] GUIDE, Design. CUDA C++ programming guide. NVIDIA, July, 2020.
>
>
> **Question 1) [I hope the authors release their implementations so that the community can benefit from them.]**
>
> We are willing to provide our implementation once the paper is accepted to contribute to the community.

---

### Author Rebuttal · Authors · 2024-08-07

We sincerely appreciate the reviewers' time and effort in providing constructive and valuable feedback on this manuscript.

In response to the reviewers' questions, we have conducted evaluations:

1. Reviewer CSWm's Question 4: We provided peak memory usage data presented in Author-Rebuttal-Table 1.

2. Reviewer CSWm's Question 5: We provided n:m sparsity data presented in Author-Rebuttal-Table 2.

3. Reviewer bd55's Weakness 2: We provided ConvNeXt-Base data presented in Author-Rebuttal-Table 3.

4. Reviewer VAUi's Weakness 1: We provided pre-processing time results in Author-Rebuttal-Table 4.

5. Reviewer VAUi's Weakness 3: We have provided edge GPU exeperiment results in Author-Rebuttal-Table 5.

---

### Decision · Program_Chairs · 2024-09-25

**Decision:**

Accept (poster)

**Comment:**

The authors propose a method of pruning depthwise-seperable convolution layers, widely found in efficient CNN architectures for vision tasks, such as EfficientNet. The proposed pruning method for depthwise seperable layers, while being a fine-grained approach, achieves real-world speedups on GPUs using a hardware-aware structured pruning approach. The results show significant real-world speedup of already efficient CNNs on GPUs, e.g. approx. 4x speedup on MobileNet-V3-Small.

The reviewer's concerns mostly centred around over-claims in the paper, and missing references, which were well-addressed overall by the authors in rebuttal. On the "the first pruning implementation on point wise/depthwise convolution" claim, I would also point out that any existing pruning literature using efficientnet/mobilenet/etc models, of which there is much, would also contradict this statement and I would recommend that they re-word the claim to make it more specific to their actual contributions (i.e. designed specifically to improve performance for these types of layers).

Overall the reviewers agreed that the work is well-written, well-motivated work with significant results.